

# The effect of complex contrast training with different training frequency on the physical performance of youth soccer players: a randomized study

Helder Barra-Moura[1,2], João Guilherme Vieira[2,3], Francisco Zacaron Werneck[4], Michal Wilk[5], Bruno Pascoalini[6], Victor Queiros[7], Gilmara Gomes de Assis[8], Marta Bichowska-Pawęska[9], Jeferson Vianna[2,3] and José Vilaça-Alves[1]

[1] Research Center in Sports Sciences, Health Sciences and Human Development (CIDESD), University of Trás-os-Montes and Alto Douro, Vila Real, Trás-os-Montes and Alto Douro, Portugal
[2] Strength Training Research Laboratory, Federal University of Juiz de Fora (UFJF), Juiz de Fora, Minas Gerais, Brazil
[3] Postgraduate Program in Physical Education, Federal University of Juiz de Fora (UFJF), Juiz de Fora, Minas Gerais, Brazil
[4] Physical Education Department, Federal University of Ouro Preto (UFOP), Ouro Preto, Minas Gerais, Brazil
[5] Institute of Sport Sciences, Jerzy Kukuczka Academy of Physical Education in Katowice, Poland
[6] Postgraduate Program in Rehabilitation Sciences and Physical-Functional Performance, Federal University of Juiz de Fora (UFJF), Juiz de Fora, Minas Gerais, Brazil
[7] Postgraduate Program in Health Sciences, Federal University of Rio Grande do Norte (UFRN), Natal, Rio Grande Norte, Brazil
[8] Postgraduate Program in Integrative Physiology, Sao Paulo State University, Araraquara, São Paulo, Brazil
[9] Gdansk University of Physical Education and Sport, Gdansk, Poland

Corresponding author
Helder Barra-Moura,
helderbarrademoura@gmail.com

## ABSTRACT

**Background:** Complex contrast training (CCT) is potentially an efficient method to improve physical abilities such as muscle strength, power output, speed, agility, are extremely important in developing soccer players of different age categories.
**Aim:** This study aimed to analyze the effects of 6 weeks of CCT program applied in different training frequency (sessions per week) on youth soccer players performance.
**Methods:** Twenty-one youth soccer players (age: 15.3 ± 1.1 years; body mass 64.9 ± 0.7 kg; height 175.4 ± 0.7 cm) were randomized into three groups: a regular pre-season training control group (G0, $n = 8$), a group with regular pre-season training plus twice-a-week CCT (G2, $n = 6$), and a group with regular pre-season training plus thrice-a-week CCT (G3, $n = 7$). The CCT consisted of soccer skills-based exercises distributed across five stations, to be performed before common regular practice during a 6-week pre-season period. The agility (505 Agility test), sprint (S5 and S15), jump (SJ and CMJ), and free kick speed (11 m from the goal) of the groups were analyzed pre- and post-CCT intervention. During a 6-week pre-season period, the players integrated CCT into their regular training sessions. Furthermore, performance variables were compared between the groups.
**Results:** A statistical difference was identified for the timepoint for the players' CMJ ($p = 0.023$; $\eta^2 = 0.343$) and the free kick speed ($p = 0.013$; $\eta^2 = 0.383$) using ANOVA. The G3 showed a significant improvement in the CMJ ($p = 0.001$) and the free kick speed ($p = 0.003$) between pre- to post-CCT test. No other significant changes in performance were observed ($p > 0.05$).

**Conclusion:** The CCT training program with a weekly frequency of 3 days per week is effective in improving free kick speed and CMJ performance in young male soccer players (U-15 and U-17 categories). CCT training programs have the potential to refine an athlete's preparation for competition. However, certain performance tests did not demonstrate substantial enhancements. Consequently, additional investigations are required to ascertain the effectiveness of CCT.

## INTRODUCTION

Stimulating muscular strength through resistance training (RT) during the biological maturation phase benefits young athletes (*Malina et al., 2015*; *McQuilliam et al., 2020*), with evidence suggesting that increased strength can improve agility, speed, and power output, crucial actions to develop athletic performance (*Cormier et al., 2020*; *Seitz et al., 2014*); resistance training effectively enhances these abilities (*Hammami et al., 2017*; *Maio Alves et al., 2010*). RT is directly associated with improvements in specific sports drills (SSD) such as change of direction, acceleration, deceleration, sprints, jumps, and kicks (*Paul, Gabbett & Nassis, 2016*; *van den Tillaar & Marques, 2009*). Building upon prior work by *Maio Alves et al. (2010)*, who introduced a similar training program, we conducted a training method known as complex contrast training (CCT). In other words, this means simultaneous training (in the same training session).

In current sports science and strength training research, strategies that incorporate both heavy and light-to-moderate loads, known as complex training (CPX), have gained prominence. However, there has been variability in the terminology used to describe these approaches. *Cormier et al. (2022)* propose a new terminology, introducing the umbrella term CPX, encompassing four distinct implementations: Contrast training (CNT), a specific type of CPX involving alternating high-load and low-load (high-velocity) exercises in a circuit performed within the same training session; Ascending training (AT), a subset of CPX in which several sets of low-load, high-velocity exercises are completed before subsequent sets of high-load exercises within the same training session; Descending training (DT), a type of CPX consisting of several sets of high-load exercises (*e.g.*, back squat) completed before the execution of subsequent sets of low-load, high-velocity exercises (*e.g.*, vertical jump) within the same training session; and French contrast training (FCNT), a subset of CPX-CNT involving exercise sets performed in the following sequence: heavy compound exercise, SSC plyometric exercise, light-to-moderate load compound exercise with high velocity, and a plyometric exercise (often assisted). These definitions were developed to bring clarity to the diverse range of training methods in the

literature. The combination of exercises using high and low loads, as well as incorporating plyometric exercises and SSD, has shown potential to improve athletes' performance in various sports (*Abade et al., 2020*; *Argus et al., 2012*; *Dodd & Alvar, 2007*; *Maio Alves et al., 2010*).

CCT strategies have demonstrated to utilize a post-activation potentiation enhancement (PAPE) in the neuromuscular system of the muscles engaged in the task. In training, the PAPE effect is achieved by employing a conditioning activity (CA) with a relatively high load (submaximal), followed by an explosive post-activation exercise with a biomechanically similar movement pattern (*Seitz & Haff, 2016*). This refers to a potential enhancement of synaptic conduction and number of recruited motor units that results in an increased rate of force development (RFD) and power output required in a specific sport skill, caused by the CA (*Docherty & Hodgson, 2007*; *Freitas et al., 2017*; *Lagrange et al., 2020*). Considering the above, a positive correlation between RFD and sports skills suggests that PAPE might improve performance in sports in which explosive skills are required (*Gołaś et al., 2017*; *Kilduff et al., 2007*). Moreover, it is important to consider the time sensitivity of PAPE. The rest interval between the CA and subsequent explosive activities is crucial for maximizing its effect. The optimal rest interval duration for inducing the desired PAPE response is still being investigated, with studies exploring a range of intervals from 0 to 21 min (*Ciocca, Tschan & Tessitore, 2021*; *Guo et al., 2022*; *Maroto-Izquierdo, Bautista & Martín Rivera, 2020*).

Furthermore, CCT has been extensively investigated with regard to training variables (*Cormier et al., 2020*); however, training frequency remains an important variable that requires further study. Recently, *Kumar et al. (2023)* showed that CCT with a different weekly frequency (2 *vs.* 3 training sessions per week) and equalized volume load had similar effects on measures of physical fitness in active adult males. Conversely, CCT performed in a group of athletes with a weekly frequency of 3 days elicited greater improvements in agility T-tests compared to CCT with a weekly frequency of 2 days. *Maio Alves et al. (2010)* found that CCT led to increased performance in S5, S15 and SJ, but the number of CCT sessions per week (1 *vs.* 2) did not affect the level of improvement in these skills. *Brito et al. (2014)* reported no significant differences in sprint and agility performances after 9 weeks of twice-per-week complex training. However, *Hammami et al. (2017)* found significant improvements in many important components of athletic performance relative to standard in-season training (*e.g.*, sprint and agility) of youth athletes (~16 years old) after 8 weeks of twice-per-week contrast training. The inconsistency in findings may be due to a high heterogeneity in the studies' methods regarding the age of the populations, exercises, training frequency, and other variables of comparison.

The aim of the present study was to analyze the effects of 6 weeks of CCT programs applied in different training frequencies (sessions per week) on the performance of youth soccer players. We compared soccer players' speed, agility, jumping ability, and free kick speed after regular training *versus* CCT performed 2 and 3 times per week during the preseason. Our hypothesis was that incorporating CCT into regular soccer training sessions would enhance the athletes' physical performance. In addition, we anticipated that
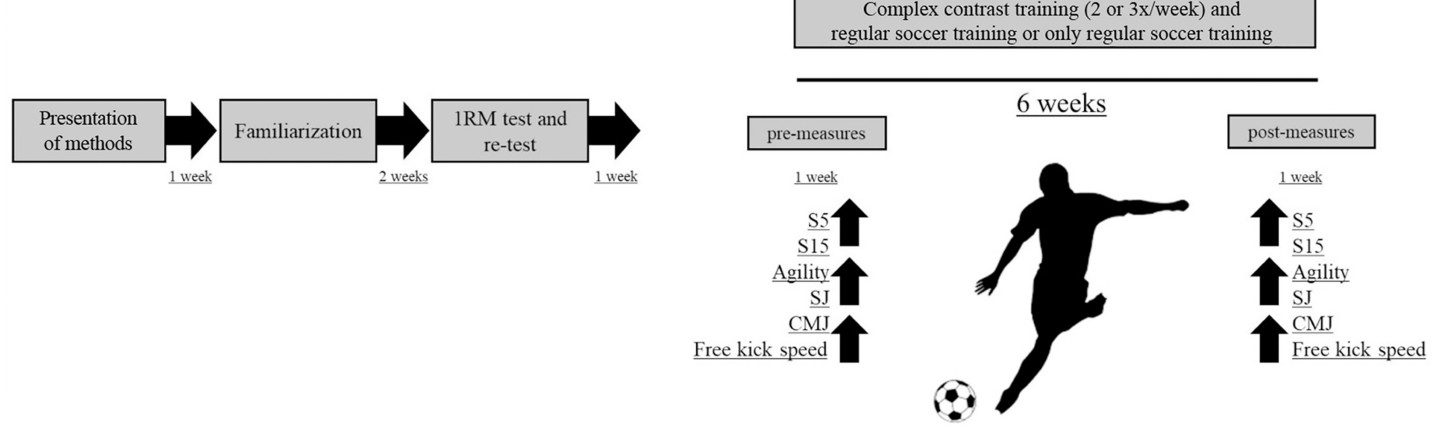

**Figure 1 Experimental design of the study.**

the degree of performance improvement would be independent of the number of CCT training sessions conducted in a weekly microcycle.

# MATERIALS AND METHODS

## Experimental approach to the problem

This study was conducted during a 12-week pre-season training period. The soccer players were randomly assigned into two experimental groups with different volumes of CCT in addition to their regular training routines, and one control group. Within- and between-factors were assessed to compare the effects of different CCT frequency on the players' sprint, agility, vertical jump, and free-kick speed. The study was conducted as: week-1; presentation of methods, week-2-3; familiarization with the CCT protocol, week-4; 1RM evaluation, week-5; performance tests, week-6-12; CCT program intervention, week-13; performance tests. Participants were instructed not to consume food or drinks with caffeine or engage in any vigorous activity other than those of the training program during the experiment (Fig. 1).

## Participants

Twenty-one soccer players (age:15.3 ± 1.1 years, body mass: 64.9 ± 0.7 kg, height: 175.4 ± 0.7 cm), participating in the Minas Gerais state Championships in Brazil (U-15 and U-17-year categories) were randomly assigned to three groups: 2 CCT sessions per week (n:6; G2), 3 CCT sessions per week (n:7; G3), and regular training as control (n:8; G0). The CCT sessions were incorporated into the regular training program. A simple randomization was conducted by an external researcher using Microsoft Excel 365 for Windows (Microsoft Corporation, Redmond, WA, USA), after pre-intervention evaluations.

## Inclusion criteria

The participants were classified as post-pubescent (adolescent) according to Tanner's scale (*Tanner, 1986*) and current medical clearance for the training protocols and research were granted. Exclusion criteria were applied to ensure the validity of the motor performance

assessment, and no participants were excluded due to the use of ergogenic substances, illnesses, pain indicators, or injuries during the experiment. Players in the U-17 category had limited previous experience in RT, with a maximum of 1 year training, while players in the U15 category had no previous experience in RT. All research protocols complied with the Declaration of Helsinki (2013) (*World Medical Association, 2013*) and were approved by the Ethics Committee of the Federal University of Juiz de Fora, Minas Gerais, Brazil (CEP: 37256314.4.0000.5147/2016). In addition, parents of all the participants signed an assent form and all participants signed an informed consent form prior to data collection.

## Anthropometric measurements

The participants' height and body weight were measured by a stadiometer (Sanny®, São Paulo, Brazil) and a mechanical scale (Filizola®, São Paulo, Brazil), while body composition was determined by focused ultrasound (Body Metrix, São Paulo, Brazil) of thoracic, midaxillary, triceps, subscapular, abdominal, suprailiac and thigh skinfolds. Body density was estimated using the *Jackson & Pollock (1978)* protocol, and percentage of body fat was then calculated using the following formula: $\frac{4.95}{\text{Body density}} - 4.5 \times 100$ (*Siri, 1993*). All measurements were performed by a single examiner.

## Performance tests

The assessments included the one-repetition maximum (1RM) test on the participants' dominant side for the hip flexion exercise. All other 1RM tests, such as leg extension, squat (guided bar), hip adduction (chair), and seated calf raise, were performed without specific consideration for the dominant side. These exercises were conducted using equipment from one brand (Strong Machine, Minas Gerais, Brazil). The evaluation followed the recommendations of the National Strength & Conditioning Association (*Haff & Triplett, 2016*). A 3-week familiarization period prior to field tests was introduced to the participants and all tests were performed on a soccer field (grass, 25–30 mm). Jumping ability was evaluated at the Strength Training Research Laboratory.

Players' agility was evaluated using the 505 Agility test by *Sheppard & Young (2006)* and the Fitness Speed Test (Speed Test Fit) using photocells (Cefise, São Paulo, Brazil). This test evaluated the ability to change direction while in motion.

Sprinting speed was evaluated at a distance of 15 m with photocells (Speed Test Fit Cefise, São Paulo, Brazil) positioned at 0.5 and 15 m. Participants were instructed to start their sprint with their foot placement positioned behind the first photocell. The timing was triggered by the interruption of the light beam when the participants crossed the first photocell. This method follows the protocol described by *Bosco, Luhtanen & Komi (1983)*. The best time out of three sprints was registered.

Jumping ability was evaluated using two different tests: the squat jump (SJ) and the countermovement jump (CMJ). For the SJ, participants started from a squat position with knees at a 90° angle, which was established using a goniometer (Staline®, Charlotte, NC, USA). In the CMJ, participants jumped from an upright position without using their arms. Both jumps were performed on a force plate (Cefise, São Paulo, Brazil) with a resolution of <0.5 kg (5N), and the peak jump height (cm) was recorded at a sampling rate of 600 Hz

**Figure 2 Exercises performed in the five stations of CCT.**

using the Vertical Jump Power software (Cefise, São Paulo, Brazil). Each participant performed three attempts for both tests, and the best jump height achieved was recorded.

Free kick speed evaluation was performed using the *Sousa, Garganta & Garganta, 2003* protocol. A soccer ball (weight 410–450 g, pressure 600–1,100 g/cm$^2$j–Penalty, São Paulo, Brazil) was placed on the grass, 11 m from the goal, and a radar gun (Stalker, Richardson, TX, USA) was positioned behind the ball for measurement of instant speed (0.013 m/s error). Players performed two free kicks with a 2-min rest interval in between. The highest ball speed was registered.

## Complex contrast training

Groups G2 and G3 had the CCT inserted at the beginning of their training routines (after warm-up) in 2 and 3 sessions per week, respectively. A specific warm-up was also included for these subjects' preparation for CCT, and CCT sessions were organized into five exercise stations including a general exercise, a multiform exercise, and a specific exercise as follows (Fig. 2):

Station 1–6 repetitions of squat (80% of 1RM) + 5-m high skipping + 5-m sprint.

Station 2–6 repetitions of seated calf raise (90% of 1RM) + 6 jumps + 3 jump headers.

Station 3–6 repetitions of knee extensions (80% of 1RM) + 6 jumps from a 60-cm bench (FIJU7944, São Paulo, Brazil) to floor + 3 jump headers from the bench.

Station 4–6 repetitions of hip adductions on a chair (80% of 1RM) + 6 jumping jacks + 5-m change of direction.

Station 5–6 repetitions of unilateral hip flexion (85% of 1RM) + 6 kicking simulations + 3 × 11-m free kicks.

After every 2 weeks of CCT training, 5% of the 1RM load was added.

## Statistical analyses

Gaussian distribution and homogeneity were verified by Shapiro-Wilk's and Levene's test, respectively, and presented in mean and standard deviation. Categorical variables were presented in relative and absolute frequencies. Two-way mixed ANOVA was used to compare the effect of the two CCT protocols (between factors-3 groups, and within factors-2 time points). One-way ANOVA was used for age comparison. Sphericity of data was evaluated by Mauchly's test, and degrees of freedom were adjusted using the Greenhouse-Geisser correction. Major effects and/or significant interactions were analyzed by multiple comparisons, with Sidak's *post hoc*. To test for differences in the proportion of maturational stages in each group, Fisher's exact test was used using contingency tables. Effect size was calculated by eta squared ($\eta^2$), with the magnitude considered as: small-$\eta^2 = 0.01$; moderate-$\eta^2 = 0.06$; and large-$\eta^2 = 0.15$ (*Cohen, 1988*). All analyses were performed in the statistical software SPSS version 27.0 (IBM Corp., Armonk, NY, USA) and the figures created in GraphPad software (Prism 8.0.1, San Diego, CA, USA) adopted a significance level of 5%.

## RESULTS

Twenty-one players completed the study (G0 = 8; G2 = 6; and G3 = 7). No differences in time point *vs*. group were found for body mass [$F_{(2,18)} = 0.988$; $p = 0.392$; $\eta^2 = 0.099$], height [$F_{(2,18)} = 0.158$; $p = 0.855$; $\eta^2 = 0.017$], BMI [$F_{(2,18)} = 1.184$; $p = 0.329$; $\eta^2 = 0.116$], and body fat percentage [$F_{(2,18)} = 1.882$; $p = 0.181$ $\eta^2 = 0.173$]; or between the groups' age [$F_{(2,18)} = 0.305$; $p = 0.741$], body mass [$F_{(2,18)} = 0.065$; $p = 0.938$], weight [$F_{(2,18)} = 0.861$; $p = 0.439$], BMI [$F_{(2,18)} = 0.230$; $p = 0.797$], and body fat percentage [$F_{(2,18)} = 1.418$; $p = 0.268$]. Fourteen participants (67%) were at maturational stage four in Tanner's scale (*Tanner, 1986*). Fisher's test did not identify a significant difference in participants' maturational stages ($p = 0.480$). Participants' characteristics are presented in Table 1.

No group-time interaction was found for S5 [$F_{(2,18)} = 0.058$; $p = 0.944$; $\eta^2 = 0.06$], S15 [$F_{(2,18)} = 0.732$; $p = 0.495$; $\eta^2 = 0.075$], agility [$F_{(2,18)} = 0.684$; $p = 0.517$; $\eta^2 = 0.071$] and SJ [$F_{(2,18)} = 1.544$; $p = 0.241$; $\eta^2 = 0.146$]. For the free kick [$F_{(2,18)} = 5.598$; $p = 0.013$; $\eta^2 = 0.383$] and the CMJ [$F_{(2,18)} = 4.701$; $p = 0.023$; $\eta^2 = 0.343$] a significant group-time interaction was found.

No effect of time point was found for S5 [$F_{(1,18)} = 0.005$; $p = 0.943$; $\eta^2 = 0.00$], S15 [$F_{(1,18)} = 0.056$; $p = 0.816$; $\eta^2 = 0.003$], agility [$F_{(1,18)} = 2.804$; $p = 0.111$; $\eta^2 = 0.135$], free

**Table 1 Anthropometric characteristics of the groups.**

|  | G0 (n = 8) | G2 (n = 6) | G3 (n = 7) | p (difference between groups) | $\eta^2$ |
|---|---|---|---|---|---|
| Age (years) | 15.13 ± 1.36 | 15.33 ± 1.03 | 15.57 ± 0.79 | 0.741 | 0.033 |
| Body mass (kg) |  |  |  |  |  |
| Pre | 63.43 ± 12.71 | 64.37 ± 12.05 | 64.99 ± 7.89 |  |  |
| Post | 63.60 ± 11.81 | 65.62 ± 11.98 | 65.86 ± 6.97 |  |  |
| MD (SD) | 0.17 ± 1.79 | 1.25 ± 1.26 | 0.87 ± 1.15 |  |  |
| p (pre vs. post) | 0.739 | 0.051 | 0.132 | 0.938 | 0.007 |
| Height (cm) |  |  |  |  |  |
| Pre | 171.38 ± 9.48 | 176.33 ± 6.67 | 177.21 ± 10.78 |  |  |
| Post | 172.21 ± 8.96 | 176.97 ± 6.48 | 177.94 ± 10.88 |  |  |
| MD (SD) | 0.84 ± 0.79 | 0.63 ± 0.55 | 0.73 ± 0.64 |  |  |
| p (pre vs. post) | 0.003 | 0.035 | 0.011 | 0.439 | 0.087 |
| BMI (kg/m²) |  |  |  |  |  |
| Pre | 21.40 ± 2.28 | 20.58 ± 2.66 | 20.66 ± 1.19 |  |  |
| Post | 21.30 ± 2.25 | 20.83 ± 2.59 | 20.80 ± 1.13 |  |  |
| MD (SD) | −0.11 ± 0.49 | 0.25 ± 0.37 | 0.13 ± 0.45 |  |  |
| p (pre vs. post) | 0.515 | 0.183 | 0.440 | 0.797 | 0.025 |
| Body fat (%) |  |  |  |  |  |
| Pre | 9.93 ± 6.93 | 6.67 ± 4.56 | 5.25 ± 1.61 |  |  |
| Post | 9.75 ± 5.42 | 7.66 ± 4.67 | 6.60 ± 1.70 |  |  |
| MD (SD) | −0.19 ± 2.27 | 0.98 ± 1.05 | 1.35 ± 0.89 |  |  |
| p (pre vs. post) | 0.746 | 0.151 | 0.039 | 0.268 | 0.136 |

Note:
G0, control group; G2, experimental group (2x/week); G3, experimental group (3x/week); n, number of participants per group; p, probability of a chance randomly generating data or something that is equal or more unusual; η2, eta squared; kg, kilograms; MD, mean difference; SD, standard deviation; pre, before intervention; post, after intervention; cm, centimeters; kg/m2, body mass divided by height squared; %, relative values.

kick [$F_{(1,18)}$ = 3.657; $p$ = 0.072; $\eta^2$ = 0.169], and SJ [$F_{(1,18)}$ = 2.022; $p$ = 0.172; $\eta^2$ = 0.101], however, there was an effect of time point for CMJ [$F_{(1,18)}$ = 10.303; $p$ = 0.005; $\eta^2$ = 0.364]. Pre- and post-CCT analyses revealed a statistically significant improvement, only in G3, for the CMJ ($p$ = 0.001–Fig. 3).

There was no statistically significant difference between the groups for S5 [$F_{(2,18)}$ = 0.381; $p$ = 0.689; $\eta^2$ = 0.041], S15 [$F_{(2,18)}$ = 0.625; $p$ = 0.547; $\eta^2$ = 0.065], agility [$F_{(2,18)}$ = 0.315; $p$ = 0.733; $\eta^2$ = 0.034], free kick [$F_{(2,18)}$ = 0.097; $p$ = 0.908; $\eta^2$ = 0.011], SJ [$F_{(2,18)}$ = 0.455; $p$ = 0.642; $\eta^2$ = 0.048], and CMJ [$F_{(2,18)}$ = 0.259; $p$ = 0.774; $\eta^2$ = 0.028]. Table 2 shows the difference in means (95% confidence interval) and Sidak's *post-hoc* values for the mentioned variables.

# DISCUSSION

The main finding of this study is that CCT training resulted in a significant improvement in the youth soccer players' performance, but only when three training sessions were performed (in addition to regular soccer training). Significant improvements were observed in CMJ and free kick speed. This demonstrates that improvements in players' physical abilities following CCT require a minimum amount of training frequency

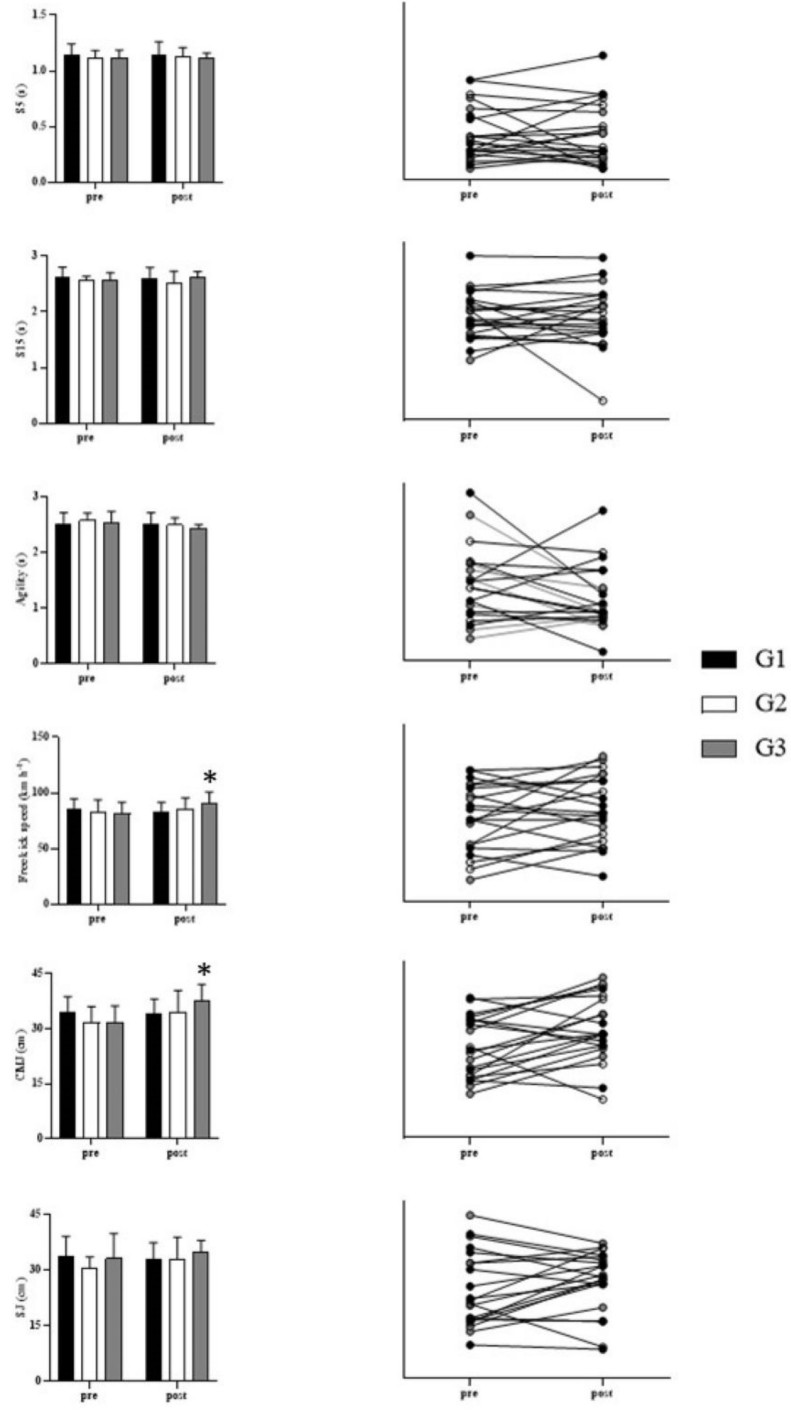

**Figure 3 Results of control group (G1) and experimental groups (G2 and G3) pre- and post-CCT.** G1, control group; G2, experimental group (2x/week); G3, experimental group (3x/week); S5, 5 meters sprint; S15, 15 meters sprint; SJ, squat jump; CMJ, countermovement jump; pre, before intervetion; post, after intervention; s, seconds; cm, centimeters; km h$^{-1}$, kilometers per hour; An asterisk (*) indicates significant difference compared with pre.

**Table 2 Comparison between groups.**

|  | Mean difference (95% confidence interval) | *Post-hoc* test |
|---|---|---|
| S5 | | |
| G0 *vs.* G2 | 0.016 [−0.112 to 0.144] | 0.984 |
| G0 *vs.* G3 | 0.031 [−0.092 to 0.154] | 0.885 |
| G2 *vs.* G3 | 0.015 [−0.117 to 0.147] | 0.987 |
| S15 | | |
| G0 *vs.* G2 | 0.107 [−0.143 to 0.357] | 0.621 |
| G0 *vs.* G3 | 0.004 [−0.236 to 0.243] | 1.000 |
| G2 *vs.* G3 | −0.103 [−0.361 to 0.154] | 0.666 |
| Agility | | |
| G0 *vs.* G2 | 0.023 [−0.183 to 0.229] | 0.988 |
| G0 *vs.* G3 | 0.094 [−0.103 to 0.292] | 0.534 |
| G2 *vs.* G3 | 0.071 [−0.141 to 0.283] | 0.773 |
| SJ | | |
| G0 *vs.* G2 | −0.263 [−6.637 to 6.112] | 0.999 |
| G0 *vs.* G3 | −2.234 [−8.343 to 3.875] | 0.724 |
| G2 *vs.* G3 | −1.971 [−8.539 to 4.596] | 0.824 |
| CMJ | | |
| G0 *vs.* G2 | −0.421 [−7.134 to 6.292] | 0.998 |
| G0 *vs.* G3 | −3.309 [−9.742 to 3.124] | 0.474 |
| G2 *vs.* G3 | −2.888 [−9.803 to 4.027] | 0.636 |
| Free kick speed | | |
| G0 *vs.* G2 | −2.792 [−16.272 to 10.688] | 0.932 |
| G0 *vs.* G3 | −7.696 [−20.615 to 5.222] | 0.351 |
| G2 *vs.* G3 | −4.905 [−18.791 to 8.982] | 0.744 |

Note:
G0, control group; G2, experimetal group (2x/week); G3, experimental group (3x/week); S5, 5 meters sprint; S15, 15 meters sprint; SJ, squat jump; CMJ, countermovement jump.

(volume). For well-trained athletes, further performance improvements may necessitate a more skill-specific approach and exercises that exhibit a dose-dependent effectiveness.

The significant increase in CMJ performance observed exclusively in players who practiced CCT 3 times per week may be attributed to a chronic enhancement in the neuromotor recruitment of muscle fibers and calcium release by the sarcoplasmic reticulum (*Tillin & Bishop, 2009*). These findings suggest that a high stimulation frequency is indispensable for eliciting substantial improvements in physical performance. Previous studies utilizing training protocols of 2–3 sessions per week have demonstrated improvements in athletes' CMJ results. However, these improvements were observed for periods longer than 6 weeks (*Cormier et al., 2020*; *Kotzamanidis et al., 2005*; *de Villarreal et al., 2015*; *Tricoli et al., 2005*). It seems that significant improvements in performance of youth soccer players require a frequency of at least 3 CCT sessions per week, or a longer CCT intervention period (*Kotzamanidis et al., 2005*; *Maio Alves et al., 2010*). Nevertheless, in this study we have demonstrated that a 6-week CCT protocol is effective in improving

CMJ in youth male soccer players from U-15 and U-17 categories, which shows to be a feasible strategy for pre-season preparations.

Although no time point effect was found for the free kick (0.072), Sidak's *post-hoc* test found a difference only for the G3 group when the different times points were compared (0.003). The sample size may not have been large enough for the primary effect to reveal the difference, but the improvement seems to have taken place. The improvement in free kick speed in the group that performed 3 CCT training sessions per week was consistent with a previous study that observed improvements in the physical fitness of young soccer players, particularly in increasing free kick efficiency (*Cavaco et al., 2014*). These improvements could be attributed to mechanisms such as improved motoneuron firing rate, increased neural coordination, and the positive transfer of muscle demands through the implementation of complex training (*Cronin, McNair & Marshall, 2001*; *Robbins, 2005*). In contrast with the findings of *Cavaco et al. (2014)*, where an improvement in athletes' free kick efficiency was found after a 6-week training program with two sessions per week, in the present study the group that included 2 CCT training sessions per week did not show such an improvement. This difference in outcomes may be attributed to variations in the protocols used, such as the specific training method employed and the number of stations utilized by *Cavaco et al. (2014)*, which may have impacted the results differently. Before the preseason period, all athletes underwent a 2-week familiarization phase with RT, followed by 6 weeks of CCT alongside regular soccer training for the experimental groups. The control group continued with RT alongside their regular soccer training. It is worth noting that there was no sudden increase in training loads during the preseason period, which could explain the observed improvements. The training programs were designed to ensure consistency and controlled progress throughout the study.

In terms of squat jump (SJ) performance, no improvement was observed in any of the studied groups following the training intervention. When comparing our study to previous studies on CCT, *Maio Alves et al. (2010)* implemented an 8-week protocol comprising 2 weeks of RT adaptation followed by a 6-week CCT. They reported improvements in SJ performance that were independent of the training frequency (1 *vs.* 2 times per week). In another experiment, *Kotzamanidis et al. (2005)* observed changes in SJ after 13 weeks of training. The absence of SJ improvement in our study may be attributed to various factors, including the need for the nervous system to adapt to control and transfer additional force after resistance training, as reported by *Bobbert & Van Soest (1994)*. The lack of SJ improvement in our study could also be due to insufficient training volume or adaptations in the muscle-tendon unit. Considering these factors, the lack of SJ improvement may be influenced by learning effects, training frequency, and rest intervals. The development of SJ performance is intricately linked to the frequency of resistance training sessions (*Hoffman et al., 1990*). Our study, which varied training frequency (2 *vs.* 3 times per week), may not have sufficient training volume to provide SJ improvements, despite previous findings suggesting the positive impact of combining resistance training with running performance on SJ (*Baker, 1996*; *Fatouros et al., 2000*). While speed training was included, SJ results did not improve, possibly due to inadequate adaptations in the muscle-tendon unit (*Young, Wilson & Byrne, 1999*), which are increased by resistance training (*Kubo, Kanehisa &*

*Fukunaga, 2002*). Therefore, lack of SJ improvement observed in our study may be influenced by several factors, including the learning effect, training frequency, and rest intervals utilized.

The present study showed no improvement in agility levels after 6 weeks of CCT training when compared to the group that did not undergo CCT, aligning with previous findings (*Maio Alves et al., 2010*; *Tricoli et al., 2005*). Despite incorporating recommended exercises that simulate directional changes in agility tests (*Maio Alves et al., 2010*), the complexity of agility tasks suggests that motor control factors have a greater impact on agility performance than muscle strength or power capacity (*Tricoli et al., 2005*). In contrast, *García-Pinillos et al. (2014)* observed improvements in agility among youth soccer players after a 12-week CCT protocol, without concurrent changes in body composition. Their study emphasized the influence of various factors such as training status, age, gender, testing methods, and program characteristics on agility outcomes. However, comparing agility gains across studies is challenging due to the diversity of evaluation methods used. Considering the complexity of agility tasks and the interplay of factors like age, training status, and testing methods, designing tailored training programs that account for individual characteristics becomes crucial for optimizing agility enhancement. Further research is warranted to explore effective strategies for improving agility in diverse populations.

In our study, consistent with various previous research (*Brito et al., 2014*; *Hammami et al., 2017*; *Kotzamanidis et al., 2005*; *Maio Alves et al., 2010*), no improvements in sprint performance (S5 and S15) were shown among any of the groups. In the study by *Brito et al. (2014)*, an improvement in the S20 sprint was observed after 9 weeks of CCT, albeit with a longer training program compared to our study. Additionally, *Maio Alves et al. (2010)* reported reductions in sprint performances (S5 and S15), but the participants in the study had an average age of 17.4 years, which is higher than the average age (15.3) of the participants in our study. During maturational stage five (post-PHV), athletes will experience an increase in testosterone and growth hormones, which leads to improved muscle strength (*Costa et al., 2021*; *Till et al., 2014*) and peak velocity (*Fernández-Galván et al., 2022*).

*Cavaco et al. (2014)* in a study with a similar age and protocol duration to ours, also found no changes in S15 sprint performance among youth athletes. Notably, a substantial portion of our sample (67%) consisted of youth soccer players in maturational stage four (on the Tanner scale), suggesting that neuromuscular maturation might be a contributing factor in sprint performance. Factors such as neural function, muscle stiffness, testosterone concentration, and muscle strength can influence the impact of maturational stage on performance (*Costa et al., 2021*; *Mendez-Villanueva et al., 2011*). *Fernández-Galván et al. (2022)* propose that lower sprinting performance in youth athletes at maturation stage four could be attributed to a decrease in relative strength due to increased body mass and height, as previously reported (*Comfort et al., 2014*). Another possible explanation is that young athletes in maturational stage four may experience a phase of "motor clumsiness" as their motor coordination is disrupted by torso and limb growth. These factors highlight the potential influence of maturational stage on sprint performance.

A limitation of our study is that the CCT protocol was implemented during the preseason, potentially introducing methodological conflicts. It might be advisable to prioritize CCT interventions during the preparatory season and possibly segment players by position (*e.g.*, defenders, forwards, and midfielders). Another limitation relates to sample size; our study included a modest group of 21 players, potentially limiting the generalizability of the findings. Additionally, we did not conduct a sample size calculation to determine the necessary number of participants. Notably, we lack the 1 RM values for the players in each group. It is worth noting that PAPE is influenced by training background and strength/power levels. Therefore, future research should aim to include larger and more diverse samples to enhance the external validity of results.

## CONCLUSIONS

The implementation of CCT in regular soccer training for youth athletes, in a regime of three sessions per week for 6 weeks, was effective in improving the athlete's CMJ and free kick speed. However, this intervention did not yield significant changes in other aspects, highlighting its limitations. Notably, the G2 volume exhibited ineffectiveness in producing desired outcomes. While further research is needed to explore optimal training protocols for enhancing other abilities such as agility and sprint performance, it is important to consider factors beyond training volume, such as maturational stage, which may play a role in reducing the dose-response and frequency while increasing the effectiveness of the intervention.

## ACKNOWLEDGEMENTS

We would like to thank all the athletes for their engagement and effort in helping us in this very important research.

## ADDITIONAL INFORMATION AND DECLARATIONS.

### Funding
The authors received no funding for this work.

### Competing Interests
The authors declare that they have no competing interests.

### Author Contributions
- Helder Barra-Moura conceived and designed the experiments, performed the experiments, analyzed the data, prepared figures and/or tables, authored or reviewed drafts of the article, and approved the final draft.
- João Guilherme Vieira conceived and designed the experiments, performed the experiments, prepared figures and/or tables, and approved the final draft.
- Francisco Zacaron Werneck analyzed the data, prepared figures and/or tables, and approved the final draft.
- Michal Wilk analyzed the data, authored or reviewed drafts of the article, and approved the final draft.

- Bruno Pascoalini performed the experiments, authored or reviewed drafts of the article, and approved the final draft.
- Victor Queiros analyzed the data, authored or reviewed drafts of the article, and approved the final draft.
- Gilmara Gomes de Assis analyzed the data, authored or reviewed drafts of the article, and approved the final draft.
- Marta Bichowska-Pawęska analyzed the data, authored or reviewed drafts of the article, and approved the final draft.
- Jeferson Vianna conceived and designed the experiments, prepared figures and/or tables, authored or reviewed drafts of the article, and approved the final draft.
- José Vilaça-Alves conceived and designed the experiments, prepared figures and/or tables, authored or reviewed drafts of the article, and approved the final draft.

## Human Ethics

The following information was supplied relating to ethical approvals (*i.e.*, approving body and any reference numbers):

The Ethical Committee of the Federal University of Juiz de For a-Minas Gerais-Brazil approved the study (37256314.4.0000.5147/2016).

## Data Availability

The raw measurements are available in the Supplemental File.

## Supplemental Information

Supplemental information for this article can be found online at http://dx.doi.org/10.7717/peerj.17103#supplemental-information.

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
