# Peer review of "The effect of complex contrast training with different training frequency on the physical performance of youth soccer players: a randomized study"

_PeerJ, doi:10.7717/peerj.17103_

## Round 0.1 · original submission · Major Revisions

Dear Authors

The manuscript has been reviewed by two experts in the field of the study. We believe your study provides valuable input for our current understanding of applied sports psychology and volleyball performance. The reviewers have addressed several issues for the next round of revision. We would like to invite you to submit a revised version of the manuscript that addresses the points raised by the reviewers, particularly regarding the definition of mental energy and its association with volleyball performance during competitions.

We look forward to receiving your revised manuscript.

Best regards

Yung-Sheng Chen, Ph.D.
Academic Editor

Reviewer 1 ·

Basic reporting

This paper has assessed the efficacy of CCT use in young soccer players comparing control, 2xweek, and 3xweek frequency in a preseason period on the development of various specific skills. The english should be reviewed as there are many sections with ambiguous language. There is sufficient literature and use of references however the terminology of CCT should be reviewed especially in the introduction which would help with clarify throughout.

The article is well structured although some formatting should be reviewed to reflect PeerJ.

Some statements do not reflect the results from the study, thus this should be reviewed.

Experimental design

This research falls within the scope of health sciences in line with the aims and scope of the journal.

It could be made more clear how the research fills an identified gap. This should be stated more clearly in the aims.

Ethical standards are stated and well done

Mostly good details in the methodology however further clarifications are needed for the applicability of the design.

Validity of the findings

The is limited novelty of the methods however it is important to replicate CCT studies with improved methodological rigor which this study has done and would benefit the literature

Conclusions should be revised

Please see the specific comments below, I hope they can help improve the manuscript.

Additional comments

Specific Comments
Abstract
Line 39-40: Perhaps combine these two sentences into one. Eg, CCT methods are potentially an efficient method to improve physical abilities such as…. Also, I would remove that there is a “lack of empirical evidence”. There are now multiple meta-analyses showing its effectiveness.
Line 43-44: you can omit “youth soccer players” since it is repeated in the next line
Line 51: a 5% exercise load is unclear, please omit this.
Line 52: players “countermovement jump”. Also, perhaps there is a better word than moment. Perhaps timepoint or post-pre?
Lines 51: Perhaps the explanation of the specific protocol methods can be replaced by a description of the statistics performed.
57: earlier it was written CMJ, jump, and now countermovement jump. Please be specific and check throughout for consistency
Keywords: avoid if the word is already in the title.
Introduction
65: I am not sure strength is the essential component. In the text by Wisloff there are some associations between stronger players and other physical qualities. I would suggest, being more specific and describing how the different qualities apply to important events in soccer. Does it aid in injury-prevention or increase performance and why?
Lines 74-87: Please correct this section. Simply put, contrast training is a subset of complex training methodology, hence complex contrast training. There have been errors in the interpretations of their terminology, so to avoid any further, we should simply used the updated terminology outlined by Cormier et al. (2022). Please change the text accordingly
Line 88: refer to CCT, not complex and contrast training separately
Line 95: It would be worth including some information on the time-dependence of the PAPE phenomenon. I believe, Cormier, Docherty and Seitz and Haff are all goo resources for this.
Lines 106-112: check for the terminology
Line 113: Remove Regarding the above Mentioned.
Line 118: refrain from using language like “assumed”
Materials and Methods
Line 123: Isnt this formatting for JSCR? Please double check the formatting to the journal PeerJ
Line 142: remove “simple”
Line 147-150: Did all participants that began the intervention, follow-through till the end without being affected by this criteria? What was the prior RT experience of the players?
Line 174: Please include further description. What was the foot placement? Was it behind the first gait? How was it triggered? All important for reproducibility. Also details on the equipment reliability? ICC or CV?
Line 186: Police can be removed. Simply write Radar Gun.
Line 190: Complex Contrast Training. And you can remove CCT in parentheses since it has already been mentioned
Line 213: 2 timepoints?
Line 213: multiple one-way anovas were performed thus it is plural. Watch out for was and weres
Line 221: separate into two sentences after the comma
Please include the players 1RM values per group. It is difficult to make inferences on how adaptations when we don’t know. PAPE is dependent on training history and strength/power levels as well.
Results
Line 225: Please be consistent. Perhaps use timepoint instead of moment.
Discussion
Line 263: What makes these athletes well-trained? Their initial performance in pre-test or their caliber or their training history? I am not sure the findings of the study support his claim. Ensure that the discussion section reflects the results from the study.
Line 267: perhaps refer to the “chronic enhancement in the …”
Line 270: “studies where was used” dosent read well
Line 275: remove “else”
Line 278: Please clarify what was done before the pre-season period. If there was a sudden increase in soccer specific training, it would be normal to see large improvements in such actions.
Lines 296-298: This section needs some revision for clarity. The sentence on Line 298 is quite broad given what we know about strength training adaptations. Perhaps consult the work of Cormie or Haff.
Conclusions
Line 344: We cant say that greater volume will be necessary. Maybe there were other factors that could have been enhanced to reduce the dose-response and frequency and increase its potency.
Also illustrate if there are some major findings that set yours apart from all the others your have compared your findings to.

·

Basic reporting

The authors need to improve the structure of the work to improve readability among practitioners.

Experimental design

Revising the methods section will increase the replicability of the study.

Validity of the findings

Researchers need to re-check their analysis to help moving forward with providing ample discussion.

Additional comments

Ln 42-43: Revise. Set to past tense; Sentence construction error.
Ln 44-46: Revise. Separate G2 and G3….They were randomized into a regular pre-season group (G0), G0 + twice-a-week CCT (G2), or G0 + thrice-a-week CCT (G3)
Ln 46-48: move this sentence after CCT description. Just use pre and post agility, sprint… of groups were analyzed.
Ln 52-55: Double check if both main effects are present to confirm true significant difference in group x time interaction.
Ln 65-73: Please change ‘muscle strength’ to power.
Ln 74-87: Improve description of complex training – completion of a heavy exercise followed by an explosive exercise…
Ln 99-112: Just use the studies that are somehow similar to the protocol of your study.
Ln 164: Revise the tests as subheadings. Avoid italicization of the tests (e.g .Agility. The 505 Test was used..)
Ln 165-166: What do you mean by participants’ dominant side when in 1 RM testing? Single-leg squat ? Please clarify. Indicate the equipment used for other 1 RMs.
Ln 178-183: Separate description for SJ and CMJ tests.
Ln 224-256: group x time interaction? To establish the ‘true’ difference in group x time, you should see if both the main effects group and time, separately are significantly different.
Ln 259-265: Specify what specific index where the 3x/week CCT was better than G0, G2 (Again recheck your analysis).
Ln 266-277: You mentioned improvement in CMJ but the concluding sentence referred to essential skills. Re-align your statements in this paragraph.
Ln 277-288: Avoid using other studies that are not related to your work. You used CCT, try to look for other literature that exhibited improvement in sport-specific skill with similar protocol. Discuss the potential mechanisms.
Ln 289-298: Focus on explaining the potential mechanisms for non-effect. Just reference other studies instead of expounding their protocols.
Ln 299-309: Try to provide an explanation to non-difference. Maybe non-specificity of the test used in your study?
Ln 310-335: Compress the sprint result in one paragraph. What are potential factors that influence non-differences in your study? Expound here.
Ln 336-339: Add 2-3 more limitations, making it into one paragraph.
Ln 342-345: countermovement jumping ability?

---

## Round 0.2 · Minor Revisions

Dear Authors

The manuscript has been reviewed by two experts in complex and contrast training. We believe your study provides merits in field of sports sciences. The reviewers have still addressed several issues for the next round of revision. We would like to invite you to submit a revised version of the manuscript that addresses the points raised by the reviewers, particularly viewing annotated manuscript provided by reviewer 2.

We look forward to receiving your revised manuscript.

Best regards

**Language Note:** The review process has identified that the English language must be improved. PeerJ can provide language editing services - please contact us at [email protected] for pricing (be sure to provide your manuscript number and title). Alternatively, you should make your own arrangements to improve the language quality and provide details in your response letter. – PeerJ Staff

Reviewer 1 ·

Basic reporting

The english is improved but please see some of the edits suggestions bellow. Some sentences are not concise .

Further, I have suggested a reading on Complex training with the proper definitions of Complex contrast training as there appears to be confusion or inconsistency throughout.

Experimental design

Research quesiton is well defined and the changes have improved the analysis.

Validity of the findings

This is a novel appraoch and one that is important as it is common practice to integrate RT into soccer specific sessions. All necessary data was provided and conclusions are well stated. However may be improved once the terminology is updated in the introduction (otherwise it appears the methods, results, discussion are adequate),

Additional comments

Abstract
Can remove: “…, refuting our hypothesis”.
Instead of using “significant” please consider using the magnitudes of the effects eg, :”presented a small improvement..” Check this throughout the manuscript.
Introduction
Consider revising the introduction line 66-70 to the following to be more concise and clear:
“Stimulating muscular strength through resistance training (RT) during the biological maturation phase benefits youth athletes (Malina et al. 2015; McQuilliam et al. 2020), with evidence suggesting that increased strength can improve agility, speed, and power output, crucial actions to develop to enhance athletic performance (Cormier et al. 2020; Seitz et 69 al. 2014).”
SSD: please clarifiy the use of complex training “along with SSD”, does this mean concurrent training (in separate sessions or before or after a training session”?
Line 86: You already introduced CCT in prior paragraph, can just put CCT, not spelled out
Line 88: Replace “complex or contrast” with just “contrast” or remove “or”.
Consider using the terminology by Cormier et al. The explanation is overcomplicated as is. Complex training is a type of training combining high and low loads (or SSD), contrast is a subset of that training where these loads are alternated. Ascending is all low before high, and descending is all high before low. All of which may lead to differing type of potentiation depending on the structure.
See the link bellow for most up to date information.
https://pubmed.ncbi.nlm.nih.gov/35816233/
Line 168: corporal density? English translation?

·

Basic reporting

The article needs professional English assistance to improve reporting.

Experimental design

Method described are sufficient for replication. Research question are well defined.

Validity of the findings

While the data and corresponding statistical methods are sound, the manner it was presented in the discussion needs further improvement.

Additional comments

Although efforts were administered, the current manuscript still needs further work to meet the journal standards. Furthermore, I strongly recommend having your work externally proofread before submission.

---

## Round 0.3 · Minor Revisions

Dear Authors

Your manuscript has been reviewed by the reviewers. The reviewers have raised issues regarding statistical reports and English proofreading. We would like to invite you to submit a revised version of the manuscript that addresses the points raised by the reviewers.

We look forward to receiving your revised manuscript.

Best regards

**Language Note:** The Academic Editor has identified that the English language must be improved. PeerJ can provide language editing services - please contact us at [email protected] for pricing (be sure to provide your manuscript number and title). Alternatively, you should make your own arrangements to improve the language quality and provide details in your response letter. – PeerJ Staff

Reviewer 1 ·

Basic reporting

this has been sufficiently corrected

Experimental design

this has been sufficiently corrected

Validity of the findings

this has been sufficiently corrected

Additional comments

well done

·

Basic reporting

The manuscript is difficult to follow, requiring someone proficient in English for review/assistance prior to resubmission.

Experimental design

The strength of the current work lies on the robust methods employed.

Validity of the findings

Findings that depicted significant interactions need to be validated by main effects to establish 'true' difference.

Additional comments

Although the efforts of the authors are commendable in the latest revision, further improvements are needed to meet the standards of Peer J. Specifically, I would recommend assistance from a proofreader proficient in English to establish the cohesiveness of the manuscript. In addition, the statistical findings should be supported by someone familiar with 2x3 repeated measures ANOVA.

---

## Round 0.4 · accepted · Accept

Dear Authors,

I want to express my thanks for your patience and efforts to improve the quality of the manuscript. Your submission is now endorsed by three experts for acceptance of publication in PeerJ. Congratulation!!!

Thank you for submitting your interesting article to PeerJ. I look forward to receiving your future research and review articles for consideration.

Best Regards
Ph.D. Yung-Sheng Chen